# Mechanism and structural diversity of exoribonuclease-resistant RNA structures in flaviviral RNAs

Andrea MacFadden[1], Zoe O'Donoghue[1], Patricia A.G.C. Silva[2], Erich G. Chapman[1,7], René C. Olsthoorn[3], Mark G. Sterken[4,5], Gorben P. Pijlman[4], Peter J. Bredenbeek[2] & Jeffrey S. Kieft [1,6]

Flaviviruses such as Yellow fever, Dengue, West Nile, and Zika generate disease-linked viral noncoding RNAs called subgenomic flavivirus RNAs. Subgenomic flavivirus RNAs result when the 5′–3′ progression of cellular exoribonuclease Xrn1 is blocked by RNA elements called Xrn1-resistant RNAs located within the viral genome's 3′-untranslated region that operate without protein co-factors. Here, we show that Xrn1-resistant RNAs can halt diverse exoribonucleases, revealing a mechanism in which they act as general mechanical blocks that 'brace' against an enzyme's surface, presenting an unfolding problem that confounds further enzyme progression. Further, we directly demonstrate that Xrn1-resistant RNAs exist in a diverse set of flaviviruses, including some specific to insects or with no known arthropod vector. These Xrn1-resistant RNAs comprise two secondary structural classes that mirror previously reported phylogenic analysis. Our discoveries have implications for the evolution of exoribonuclease resistance, the use of Xrn1-resistant RNAs in synthetic biology, and the development of new therapies.

[1] Department of Biochemistry and Molecular Genetics, University of Colorado Denver School of Medicine, Aurora, CO 80045, USA. [2] Department of Medical Microbiology, Leiden University Medical Center, 2333 ZA Leiden, The Netherlands. [3] Leiden Institute of Chemistry, Leiden University, Einsteinweg 55, 2333CC Leiden, The Netherlands. [4] Laboratory of Virology, Wageningen University, Droevendaalsesteeg 1, 6708PB Wageningen, The Netherlands. [5] Laboratory of Nematology, Wageningen University, Droevendaalsesteeg 1, 6708PB Wageningen, The Netherlands. [6] RNA BioScience Initiative, University of Colorado Denver School of Medicine, Aurora, CO 80045, USA. [7] Present address: Department of Chemistry and Biochemistry, University of Denver, Denver, CO 80208, USA. Andrea MacFadden and Zoe O'Donoghue contributed equally to this work. Correspondence and requests for materials should be addressed to J.S.K. (email: Jeffrey.Kieft@ucdenver.edu)

Flaviviruses are single-stranded, (+)-sense RNA viruses with 10–11 kb-long genomes[1]. Infection by mosquito-borne flaviviruses (MBFVs) results in amplification of the genomic RNA (gRNA) and also production of noncoding subgenomic flaviviral RNAs (sfRNAs)[2–8]. sfRNAs accumulate to high levels, interacting with many cellular proteins to influence processes such as RNA interference, proper cellular RNA decay, the interferon response, and the process of transmission between mosquito vector and vertebrate host[9–19]. sfRNAs have been implicated in cytopathicity in cell culture and in pathogenicity in fetal mice[6,20], thus they are directly related to disease symptoms and are potential therapeutic targets.

MBFV sfRNAs are formed by partial degradation of the viral genomic RNA by cellular 5′–3′ exoribonuclease Xrn1, an important enzyme in normal RNA decay pathways that degrades 5′ monophosphorylated RNAs (Fig. 1a)[21]. MBFV genomes contain discrete RNA structures in their 3′-untranslated region (UTR) that block the progression of Xrn1. These RNA elements are sufficient to block Xrn1 without the use of accessory proteins, thus they have been assigned the name 'Xrn1-resistant RNAs' (xrRNAs)[6,13,22–27]. xrRNAs halt the enzyme at a defined location such that the viral RNA located downstream of the xrRNAs is protected from degradation. These protected RNAs are sfRNAs, and in some but not all cases multiple xrRNA structures give rise to multiple sfRNA species (Fig. 1a)[6,11,17,22–29].

Xrn1 can unwind and degrade highly structured RNAs such as picornaviral IRES elements[6] and ribosomal RNA, thus the ability of discrete RNA structures in MBFVs to block the progression of Xrn1 is surprising, and the mechanism was poorly understood. Structures of xrRNAs from Murray Valley encephalitis virus and Zika virus solved by x-ray crystallography revealed that a three-way junction and multiple pseudoknot interactions create an unusual and complex fold that requires a set of nucleotides conserved across the MBFVs[23,27]. In the fold, the 5′-end of the RNA passes through a ring-like structure (Fig. 1b), and modeling suggested that resistance occurs when this ring-like structure contacts the surface of Xrn1. However, the mechanism of how this leads to Xrn1 resistance remained speculative. It has been proposed that Xrn1's helicase function involves two alpha helices that assist in unwinding double-stranded RNA, producing 5–6 nucleotides of single-stranded RNA that span the distance from the enzyme's surface to its active site where the RNA is cleaved[30,31]. Based on this, various mechanistic ideas could be proposed: i) the conserved xrRNA structure and sequence could make specific contacts to Xrn1's surface to prevent helicase activity or block enzyme conformational changes, ii) it could somehow alter the specific catalytic mechanism of Xrn1's active site, iii) it could use nonspecific physical interactions with the enzyme, iv) it could present a general mechanical unfolding problem, or it could use some combination of these strategies.

In addition, although the formation of sfRNA in the MBFVs and tick-borne flaviviruses (TBFVs) is well-established[6,28,32], direct evidence is lacking for whether other non-arthropod-borne flaviviruses also form sfRNAs. These flaviviruses include members of the no known arthropod vector flaviviruses (NKVFVs) that infect small mammals and bats, and the insect-specific flaviviruses (ISFVs) that have no identified vertebrate host. While the sequences of MBFV xrRNAs contain a number of absolutely conserved nucleotides within a shared three-dimensional fold[6,25], the non-MBFV putative xrRNAs do not have the same sequences as those found in MBFVs[19,28,29]. If these ISFVs and NKVFVs do form sfRNAs, this raises questions of whether they arise from folded RNA structures that function without protein co-factors, and how their structures compare with the MBFV xrRNAs.

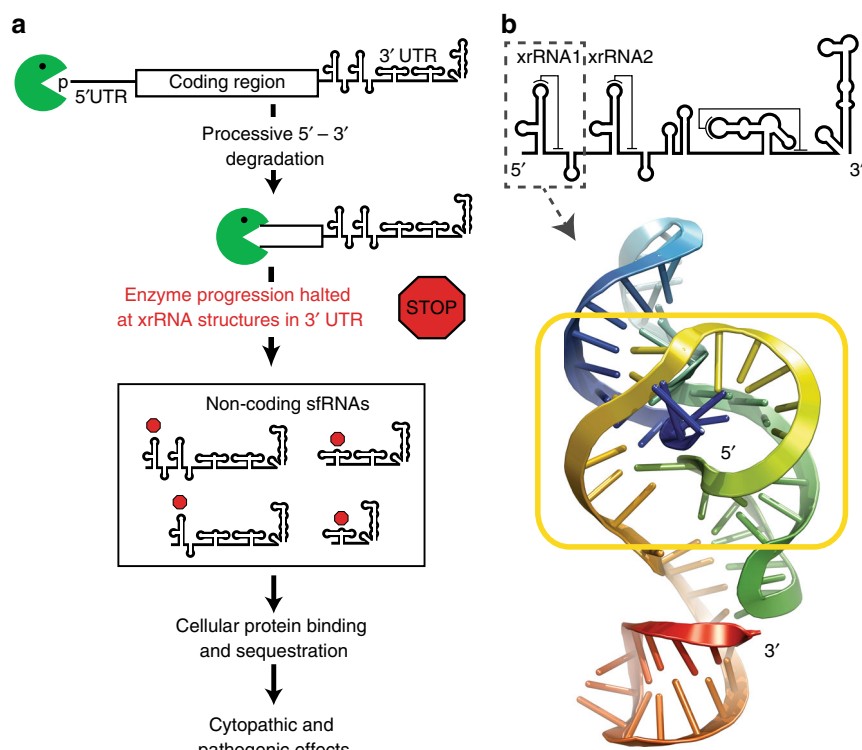

**Fig. 1** Formation of sfRNAs by Xrn1 resistance. **a** Xrn1 degrades the viral genomic RNA in a 5′–3′ direction, but halts at xrRNA structures. The resultant noncoding viral sfRNA is formed, which affects several pathways. **b** Three-dimensional structure of the upstream xrRNA from Zika virus shown in cartoon form, in rainbow. The 5′-end is blue and 3′-end is red. The fold forms a unique ring-like structure (yellow box) through which the 5′-end of the RNA passes. The location of this xrRNA in a generic MBFV 3′-UTR is shown above the structure

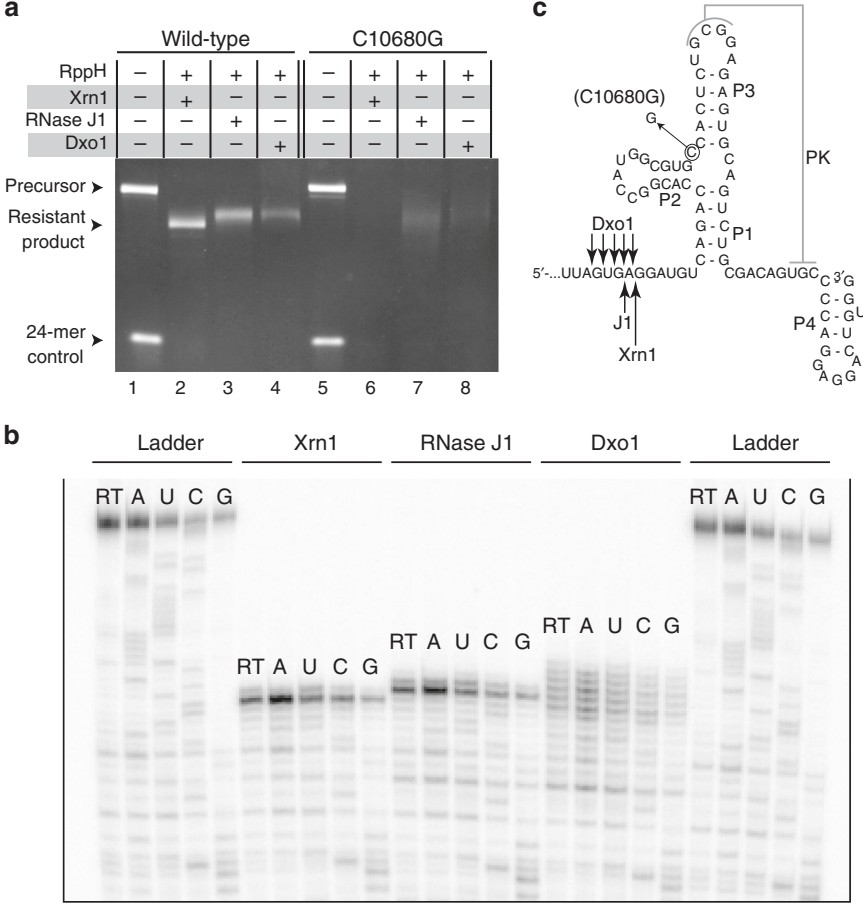

**Fig. 2** Testing the mechanism of exoribonuclease resistance. **a** Exoribonuclease resistance assay of wild-type and mutant WNV xrRNA2, using different exoribonucleases. **b** Reverse transcription mapping of the halt sites of the exoribonucleases. **c** Secondary structure of the test xrRNA with the location of the point mutation and the halt sites for all of the exoribonucleases indicated. Gels are representative of greater than three independent experiments

Here, we describe studies to address fundamental questions of the mechanism underlying Xrn1 resistance by MBFV xrRNAs and the existence and structure of xrRNAs from other flaviviruses. Using a biochemical approach, we discovered that xrRNAs can halt exoribonucleases unrelated to Xrn1 and that the nature of the interface between the RNA and the enzyme affects where the enzyme halts. This reveals that xrRNAs operate as general mechanical blocks to diverse molecular machines, with implications for the use of these RNAs in diverse contexts. Furthermore, we directly demonstrate that Xrn1 resistance can be achieved by sequences within the 3′-UTR of flavivirus RNAs outside of the MBFVs, but that in some groups the xrRNAs use a secondary structural strategy that differs from that of the MBFVs. By extension, structured exoribonuclease-resistant RNAs may be widespread, possibly conferring novel pathways for RNA maturation or regulation of mRNA decay.

## Results

**A flaviviral xrRNA can block diverse exoribonucleases**. A key question to understand the mechanism of xrRNA function is the specificity for Xrn1; if xrRNAs can block other unrelated enzymes it eliminates the need for specific RNA–protein interactions, for conformational changes unique to Xrn1, or for other Xrn1-specific features. We therefore tested the ability of a MBFV xrRNA to block 5′–3′ exoribonucleases other than Xrn1 using our previously characterized assay (Supplementary Fig. 1), reasoning that if these enzymes degrade through the structure, then specific

Xrn1-xrRNA interactions are likely necessary for the mechanism. We chose the second of two tandem xrRNAs (xrRNA2) from the Kunjin strain of West Nile Virus (WNV$_{KUN}$) as a representative xrRNA and challenged it with pure bacterial RNase J1, a 5′–3′ exoribonuclease unrelated to Xrn1 (Fig. 2a)[33,34]. The xrRNA effectively blocked the progression of the enzyme (Fig. 2a, lane 3). We then tested an xrRNA mutated (C10680G) to alter its three-dimensional fold and ablate Xrn1 resistance[22,23,27]. RNase J1 degraded this mutant RNA (Fig. 2a, lane 7). We then challenged the xrRNA with yeast exoribonuclease Dxo1, an enzyme that removes aberrant 5′ caps and also degrades RNA in a 5′–3′ direction[35]. Similar to RNase J1, the wild-type xrRNA resisted degradation by Dxo1 but the C10680G mutant did not (Fig. 2a, lanes 4, 8). These results indicate that the xrRNA can block diverse exoribonucleases in a manner that depends on its specific fold but does not depend on an enzyme-specific mechanism of processive RNA degradation or specific enzyme-xrRNA interactions.

**Different exoribonucleases halt at different locations**. To understand similarities or differences in the interactions driving an xrRNA's ability to block diverse 5′–3′ exoribonucleases, we mapped each aforementioned enzyme's xrRNA-induced halt site using a primer extension method (Fig. 2b, c)[22]. Interestingly, RNase J1 halts a single nucleotide upstream (5′) of Xrn1's halt site. Because these exoribonucleases have narrow entrance tunnels to their active sites that only allow single-stranded RNA

to enter, the difference in halt sites likely correlates with the distance between the active site and the surface of each enzyme[33,35]. In contrast to RNase J1, the halt site of Dxo1 was somewhat poorly defined, with the enzyme appearing to 'stutter' as it interacts with the xrRNA (Fig. 2b, c). These results have mechanistic implications. They further support the idea that the xrRNA is a general mechanical block in which the ring structure braces against the enzyme's surface around the active site entrance, and the enzyme halt site reflects the distance between the surface and the active site. However, although the xrRNA-enzyme interactions are not specific to a certain enzyme, the geometry or other characteristics of the enzyme's surface affect the precision of the halting event.

**xrRNAs in MBFVs and ISFVs use a similar structural strategy.** RNA structure-dependent Xrn1 resistance in vitro has been observed by many MBFV xrRNAs and they have been characterized biochemically, functionally, and biophysically[22,25]; in contrast, the characteristics of putative xrRNAs from other flaviviruses are more mysterious. In the MBFVs, the xrRNAs exhibit a high degree of sequence conservation with one another and they almost certainly all form a similar three-dimensional fold[6,25], but the sequences of the 3′-UTRs of flaviviruses outside of the MBFVs do not contain the same sequences that are absolutely conserved in the MBFV xrRNAs.

We first explored sfRNA production by a representative ISFV, cell-fusing agent virus (CFAV), by infecting C6/36 cells. Northern blot analysis with a probe to the 3′-end of the viral RNA revealed a robust band consistent with the production of sfRNA, and reverse transcription showed the location in the 3′-UTR (Fig. 3a). To determine whether sfRNA production is owing to Xrn1 resistance, we challenged in vitro- transcribed RNA of the CFAV 3′-UTR with purified Xrn1. The 3′-UTR RNA was processed to a single shorter degradation intermediate, indicating the presence of at least one authentic xrRNA (Fig. 3b). We mapped the enzyme halt site on the in vitro-processed RNA, revealing the location of the most upstream Xrn1-resistant element (Fig. 3c). As the RNA sequence downstream of the halt site does not match that of the characterized MBFV xrRNAs, we used selective 2′-hydroxyl acylation analyzed by primer extension (SHAPE) combined with thermodynamic predictions to generate an experimentally supported secondary structural model (Supplementary Fig. 2). We also found evidence for a second similar structural element further downstream in the 3′-UTR that likely comprises a second CFAV xrRNA.

Examination of these secondary structures show that several tertiary structure interactions critical for maintaining the three-dimensional structure of the MBFV xrRNAs can also form in the CFAV xrRNAs (Fig. 3d). Specifically, there is potential for base pairing between two nucleotides at the 5′-end and two nucleotides in the three-way junction, for the formation of a long-range pseudoknot between the L3 loop and the S4 region downstream, and for a non-canonical base pair that helps define the ring around the 5′-end. This suggests that these CFAV xrRNAs fold similarly to the MBFV xrRNAs.

Despite the aforementioned similarities, several critical tertiary interactions present in the MBFVs are missing or altered in the CFAV xrRNAs, including a U•A-U base triple interaction absolutely conserved in the MBFVs. Interestingly, in the CFAV xrRNAs, the U base and A-U base pair that form the triple have been replaced by a C base and G-C base pair (Fig. 3d, e). We predict that this triple substitution allows formation of a $C^+$•G-C base triple, which is isosteric with the U•A-U and thus can structurally replace it. To test this, we generated mutant versions of the xrRNA and tested them for Xrn1 resistance in vitro

(Fig. 3f). Substitution of the G-C with an A-U in the upstream xrRNA (disrupting the putative base triple) eliminated robust Xrn1 resistance and allowed the enzyme to progress to the downstream xrRNA (Fig. 3f). When the C was replaced with a U (fully replacing the putative $C^+$•G-C with a U•A-U), resistance was recovered; this provides strong evidence for the existence of a base triple (Fig. 3f). In addition, in the MBFVs there is an absolutely conserved C in the three-way junction that makes hydrogen bonds with the backbone and with base functional groups in the P1 stem[22]. In both CFAV xrRNAs, this C is replaced by a U, but we predict this mutation is tolerated within the overall fold (Fig. 3e). We mutated this U to either a C or an A; both were able to block Xrn1 in vitro (Fig. 3f). Interestingly, the P1 stem is considerably shorter in the CFAV xrRNAs than in the MBFVs. This may be tolerated by adjustments to other parts of the RNA fold; understanding how this is achieved will require three-dimensional structural information. By aligning the sequences of predicted xrRNAs from several members of the ISFVs, conserved tertiary interaction patterns emerge (Fig. 3g). These results suggest that these ISFV xrRNAs all adopt a three-dimensional fold similar to MBFV xrRNAs and use the same topological strategy to block Xrn1.

**More divergent flaviviruses use different xrRNA structures.** In addition to the ISFVs, other flaviviruses include the TBFVs and NKVFVs. sfRNA formation has been reported in the TBFVs[6,32], and the structures that are responsible for sfRNA formation have highly conserved sequences that do not match those of the MBFVs[32]. The NKVFVs do not comprise a single phylogenetic group. Examination of at least one member, Yokose Virus (YOKV), reveals a sequence that matches the xrRNA patterns of the MBFVs[25], but many other members do not have this sequence pattern. To examine sfRNA formation in these divergent NKVFVs, we infected BHK-21J cells with Modoc virus (MODV), Montana myotis leukoencephalitis virus (MMLV), Apoi virus (ApoiV), and Rio Bravo Virus (RBV); these represent various groups of NKVFVs. Northern blot analysis with probes to the 3′-ends of the viruses revealed the presence of sfRNAs in all (Fig. 4a). Reverse transcription mapping showed that the 5′-end of these sfRNAs was within the 3′-UTR of the viral RNA (Fig. 4b).

To determine whether these sfRNAs result from Xrn1 resistance and operate without protein co-factors, we chose representative members of the NKVFV and TBFV virus groups and challenged their 3′-UTR RNAs with Xrn1 (Fig. 4c, d). Specifically, we used in vitro-transcribed RNA comprising the full 3′-UTRs of MODV, MMLV, and TBEV. These 3′-UTRs were processed to a shorter degradation intermediate by Xrn1, indicating the presence of at least one authentic xrRNA. Interestingly, the 3′-UTR of TBEV gave rise to multiple degradation intermediates, unusual as generally we observe only a single degradation intermediate in vitro, corresponding to the most upstream xrRNA even when multiple xrRNAs exist in the 3′-UTR. We mapped the location of all Xrn1 halt sites; as with the MBFV xrRNAs, the halt sites are precise to within one or two nucleotides and their location matches those mapped on RNAs from infected cells (Fig. 4b, e, f).

RNA sequences downstream of the halt sites in the TBFVs and NKVFVs do not contain the conserved sequences observed in MBFV xrRNAs, thus they may use a different structural strategy to halt Xrn1. Indeed, the secondary structures of these RNAs have been predicted[28,29,32], but they have not been experimentally examined. We therefore used SHAPE combined with thermodynamic and phylogenetic analysis to generate secondary structure models of the MODV, MMLV, and TBEV xrRNAs

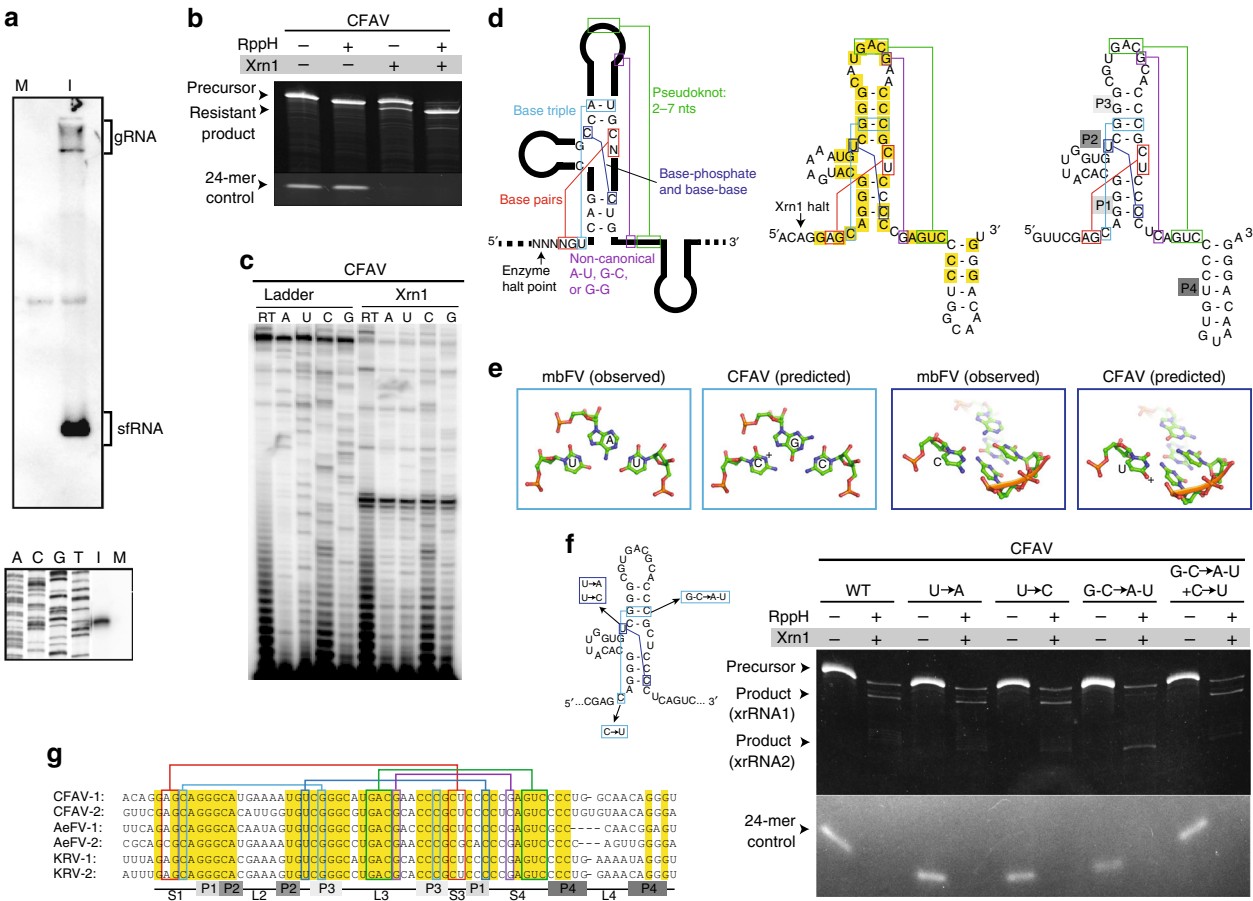

**Fig. 3** Characterization of Xrn1-resistant structures in the 3′-UTR of CFAV. **a** Above: Northern blot analysis of total RNA from CFAV-infected C6/36 cells, using a probe to the viral 3′-UTR. Below: reverse transcription mapping of the 5′-end of the CFAV sfRNA using RNA from infected cells. M = mock infected, I = infected. **b** In vitro Xrn1 resistance assay using the full 3′-UTR of CFAV. RppH = RNA 5′ Pyrophosphohydrolase (RppH). **c** Reverse transcription mapping the Xrn1 halt site with RNA from **b**. **d** Left: Schematic of a typical MBFV xrRNA with the conserved sequence, secondary, and tertiary structures indicated. Center: secondary structure model of CFAV xrRNA1. Predicted tertiary interactions analogous to those in the MBFV are shown and the Xrn1 halt site is indicated. Right: Secondary structure model of CFAV xrRNA2. Predicted tertiary interactions are shown. The location of secondary structure elements is shown in gray boxes. **e** Comparison of known tertiary interactions in the MBFV xrRNAs with predicted analogous interactions in CFAV xrRNAs, with sequence variation. **f** Testing of predicted interactions in the CFAV xrRNA1 that differ compared to the MBFVs. **g** Sequence alignments of multiple ISFVs. Absolutely conserved nucleotides are in yellow. Colored boxes and lines denote the tertiary interactions shown in **d**, with colors to match. The location of secondary structure elements shown in **d** are indicated below. Accession numbers for sequences used in **b**–**d** and **f**: CFAV, NC_001564; Aedes flavivirus (AeFV), NC_012932; Kamiti River virus (KRV), NC_005064. Gels are representative of greater than three independent experiments

(Fig. 5a–c). In all three RNAs, the mapped halt sites are followed by a similar secondary structure (two copies of this secondary structure are seen in TBEV); the similarity of these secondary structures across viral species increases the confidence that the models are correct. The secondary structures of these divergent xrRNAs contain a three-way junction, but one that differs from those found in the MBFVs and ISFVs. Whereas the three-dimensional structures of MBFVs revealed that these three-way junctions are of the "C" family, assignment of the family of these TBFV and NKVFV xrRNAs is ambiguous and thus the stacking arrangements of the emerging helices are difficult to predict[36,37]. In addition, although in the MBFVs the halt site is 5–6 nucleotides before the base of the P1 stem in a single-stranded region, in the MODV, MMLV, and both TBEV xrRNAs, the halt site corresponds to a bulge in a stem, and there is no analogous distance between the halt site and the junction (Fig. 5a–c). Overall, these xrRNAs appear to comprise a different secondary structural class compared to the MBFV xrRNAs (Fig. 6a).

An interesting putative interaction in the MODV, MMLV, and both TBEV xrRNAs is long-range base pairing between an apical loop and a sequence ~30–40 nucleotides downstream (Fig. 5a–c, green sequences and arrows)[28,32]. The MBFV xrRNAs also contain a long-range base pairing interaction that forms a functionally critical pseudoknot (green, Fig. 3d)[22,24,26], but in the MBFV xrRNAs the length of intervening sequences is shorter and mostly involved in secondary and tertiary structure[6,25]. Hence, it is not clear if the long-range pairing in the MODV, MMLV, and both TBEV xrRNAs is analogous to that observed in the MBFV xrRNAs. In the MODV xrRNA, the putative base pairing is between a 5′-UGAC-3′ sequence in the apical loop and a 5′-GUCA-3′ sequence 37 nucleotides downstream, therefore we tested Xrn1 resistance of an RNA that ended immediately after the 5′-GUCA-3′ (Fig. 5c). The wild-type version of this RNA was able to block Xrn1, indicating that although the 5′-GUCA-3′ appears to be located within a downstream stem–loop structure, this downstream stem–loop is not necessary for Xrn1 resistance.

To directly test the functional importance of the putative pseudoknot, we individually mutated two of the four bases in the apical loop or downstream pairing region and tested for resistance (Fig. 5d; dsPK-mut and usPK-mut). Both mutants lost

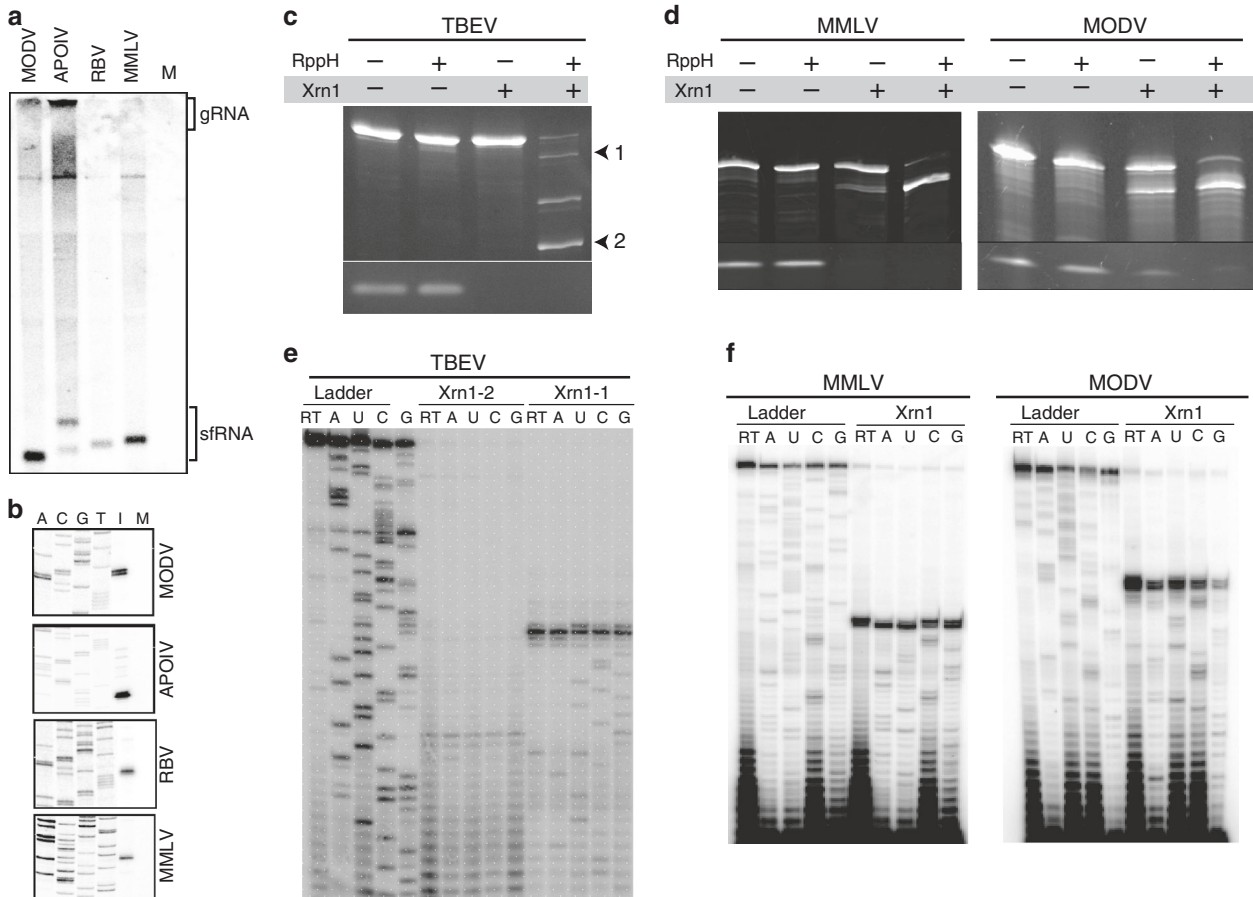

**Fig. 4** Examination of sfRNAs and direct Xrn1 resistance in TBFVs and some NKVFVs. **a** Northern blot analysis of total RNA from BHK-21J cells infected with various viruses, using a probe of the viral 3′-UTR. M = mock infected. **b** Reverse transcription mapping of the 5′-end of the sfRNAs from **a**. M = mock infected, I = infected. **c** In vitro Xrn1 resistance assay with the 3′-UTR of TBEV. RppH = RNA 5′ Pyrophosphohydrolase (RppH). 1 and 2 denote two Xrn1-resistant products. **d** Xrn1 resistance assay with the 3′-UTRs of MMLV and MODV. **e** Reverse transcription mapping of the Xrn1 halt sites in the 3′-UTR of TBEV. **f** Reverse transcription mapping of the Xrn1 halt sites in the 3′-UTR of MMLV and MODV. Accession numbers for RNA sequences used in **c**–**f**: TBEV, NC_001672; MMLV, MC_004119; MODV, NC_003635. Gels are representative of greater than three independent experiments

the ability to block Xrn1, but the presence of both mutants in the same RNA (compensatory) restored activity (Fig. 5d). We then made RNAs in which the 3′-end was truncated 10 nucleotides at a time to remove the 5′-GUCA-3′ sequence and shorten the 3′-end (Fig. 5d; mutants −10, −20, −30); none could block Xrn1. Owing to the lack of an infectious cDNA we verified the importance of the long-range interaction by transfecting BHK-21J cells with a Sindbis virus replicon RNA containing either the 3′-UTR of MODV or MMLV; analysis of the resultant RNAs by Northern blot revealed the production of sfRNAs from replicons containing the WT 3′-UTRs, but a loss of sfRNA production when the long-range interaction was eliminated (Supplementary Fig. 3). Finally, we made internal deletions to the single-stranded RNA located upstream of the 5′-GUCA-3′ sequence; these mutants retained the potential for base pairing but changed the sequential distance between the putative paired nucleotides (Fig. 5e; mutants Δ15, Δ10, Δ5). When challenged with pure Xrn1 in vitro, none of these mutants blocked the exoribonuclease. These results indicate that the putative long-range base pairing sequence is important for function and that the functional 3′-end of the xrRNA lies immediately after this sequence. Furthermore, the intervening RNA between the putative pairing sequences, whereas predicted to be single-stranded, is also important, although it is not clear whether this is to maintain a certain length or if there are specific sequence requirements.

Using information from the secondary structures of the TBEV, MODV, and MMLV RNAs, we aligned the sequences of likely xrRNAs from additional TBFV and NKVFVs (Supplementary Fig. 4 and 5). Although there is substantial sequence variation, all show patterns that are consistent with the ability of the RNA within each group to form similar secondary structures and long-range interactions. Interestingly, the intervening RNA between the putative long-range sequences contains conserved sequences, suggesting it has some important role despite it appearing to be single-stranded. Overall, these results suggest that a common structure enables Xrn1 resistance (and thus sfRNA production) in these 3′-UTRs, but that the secondary structure differs from those found in the MBFVs and ISFVs (Fig. 6a).

## Discussion

The RNA genomes of MBFVs harbor Xrn1-resistant structures in their 3′-UTRs, which block the progression of the exoribonuclease to produce disease-related noncoding RNAs[6,11]. This programmed resistance to Xrn1 depends on a specific folded three-dimensional RNA structure that is highly conserved across the MBFVs[22,23,25,27]. In this work, we interrogated the mechanism for this RNA structure-driven process and we identified and characterized xrRNAs in other flaviviruses. These discoveries have implications for the role of xrRNAs in diverse

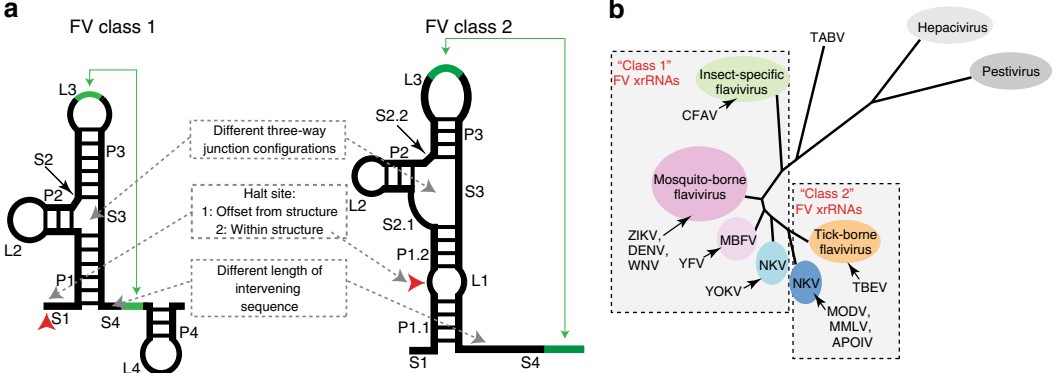

**Fig. 5** Characterization of NKVFV and TBFV xrRNAs. Secondary structures of relevant parts of the 3′-UTRs of **a** TBEV, **b** MMLV, and **c** MODV. The mapped Xrn1 halt site is indicated with black arrows. Nucleotides are shaded to indicate their normalized SHAPE reactivity. **d** Xrn1 resistance assay using wild-type, mutant, and truncated MODV xrRNA. The diagram at the top shows the mutations made. **e** Same as **d**, with deletions to the single-stranded RNA upstream of the 3′ part of the putative long-range interaction. Accession numbers for RNA sequences used: TBEV, NC_001672; MMLV, MC_004119; MODV, NC_003635. Gels are representative of greater than three independent experiments

**Fig. 6** Correlation of classes of flavivirus xrRNA with viral phylogeny. **a** Cartoon representation of the basic secondary structure of two types of flavivirus xrRNAs. The secondary structural elements in each are labeled with a proposed nomenclature. Key differences between the two are described. Green lines and arrows indicate long-range base pairing, red arrows indicate the halt point for Xrn1. **b** The phylogenetic tree of the *Flaviviridae* is an adaptation from Vlachakis et al.[52]. Shaded boxes indicate the clades of viruses that have similar classes of xrRNA, based on our structural and functional analysis. There are two clades of NKVFVs; one group has 'Class 1' xrRNAs that match the mosquito-borne viruses, and the other has 'Class 2' xrRNAs that align with the tick-borne viruses. Abbreviations not used in main text; ZIKV: Zika virus; POWV: Powassan virus; TABV: Tamana bat virus

viruses and hosts, the evolution of xrRNAs, the possibilities of analogous RNAs in other contexts, and the use of xrRNAs in nanotechnology or synthetic biology.

Previously solved structures allowed many new questions to be proposed regarding the mechanism of xrRNA resistance, which until now were untested. A central mystery was whether xrRNAs were evolved to interact with and block Xrn1 exclusively, which would suggest that specific interactions are needed between the xrRNA and the enzyme to elicit Xrn1-specific effects. In contrast, if xrRNAs could halt diverse 5′–3′ exoribonucleases, this would reveal that the structure is a general mechanical block. Consistent with the latter idea, we show that a representative MBFV xrRNA is also capable of halting RNase J1 and Dxo1, directly demonstrating the general nature of the ability to halt diverse exoribonucleases and arguing against the importance of specific contacts between the xrRNA and amino acids on the enzyme's surface. The fact that RNase J1 and Dxo1 are enzymes that xrRNAs never naturally encounter, and that xrRNAs can block their progression, are strong evidence in support of this. Also, the fact that these enzymes stop at different locations relative to the xrRNA structure is consistent with the idea that the structure braces against the surface to stop exoribonuclease progression, and this difference is a measure of the distance between the surface and the active site in the enzyme's interior[23,31]. Last, although specific contacts between the xrRNA and the enzyme surface are not critical, our data suggest that the overall shape of the surface and the type of helicase activity may affect xrRNA efficiency.

Combined with previous data, the mechanism that emerges is one in which xrRNAs resist progression of exoribonucleases using a specific three-dimensional RNA fold that braces against the surface of the enzyme and creates a general mechanical block to continued enzyme movement. Several existing pieces of evidence fit this mechanistic model. First, although the MBFV xrRNAs block Xrn1 approaching from the 5′ side, the viral RNA-dependent RNA polymerase can pass through from the 3′ side, as can the reverse transcriptase used in our experiments. Although the nature of these enzymes' helicase activity may differ from Xrn1's, this observation is consistent with resistance owing to an encounter with the 5′ side of the structure. Also, while the global structural stability of the xrRNA may play a role in resistance, Xrn1 passes through many very stable structured RNAs, suggesting thermodynamic stability is not the sole source of resistance[38,39]. Finally, the three-dimensional fold of the MBFV xrRNAs contains a unique ring-like structure through which the 5′-end passes, a feature not previously seen in other RNA structures[23,27]. Modeling shows that this ring contacts the enzyme's surface and encircles the entrance to the active site while emerging single-stranded RNA extends into the active site[23,27]. It is important to differentiate this 'mechanical block' model from one in which the overall thermodynamic stability of the fold (predicted or measured) is the primary determinant of resistance; rather than thermodynamic stability, resistance is conferred by a specific RNA topology that the enzyme encounters as it approaches from the 5′ side, and this structure presents a specific barrier to progression.

The mechanism supported by the data has several implications for the function of these xrRNAs in flavivirus infection. The ability of the MBFV xrRNAs to resist diverse exoribonucleases may assist them in operating in a wide range of hosts and vectors. In other words, sequence differences in Xrn1 from different species are unlikely to result in an enzyme that is able to degrade through an xrRNA. Although viral tropism is dependent on many factors, at least the xrRNA is likely to be functional in a very wide range of species[17,18]. Likewise, versions of Xrn1 capable of overcoming xrRNAs are unlikely to arise through a specific mutation within a species, eliminating one host-centered mechanism of evolving resistance to the virus. The ability to block diverse exoribonucleases also confers the potential to use processive 5′–3′ exoribonucleases other than Xrn1 to produce sfRNAs. Indeed, Xrn1 knockdown experiments in Zika virus suggested the presence of redundancy in the 5′–3′ decay machinery that could be exploited by xrRNAs that block diverse exoribonucleases[27].

The ability to halt Xrn1 and potentially other cellular exoribonucleases is shared across the flavivirus genus, although the RNA structural determinants of this function have diverged. Specifically, comparing the structural characteristics of the Xrn1-resistant elements from some members of the NKVFVs and the TBFVs with those from the MBFVs reveals obvious differences. Although the experimentally-supported secondary structure models of these diverse xrRNAs all contain three-way helical junctions, they are only superficially similar when compared with known RNA three-way junctions[37]. In addition, the distance (in terms of sequence length) between the sequences involved in a putative long-range base pairing interaction is longer than is typical for the MBFVs[25]. Likewise, the halt point for the exoribonuclease in the tested NKVFV and TBFV xrRNAs lies within a helical element and only 5–6 nucleotides upstream of the junction. This is potentially significant because as the enzyme approaches, it will unwind these helical elements, with the effect of further increasing the amount of single-stranded RNA between the structure and the long-range base pairing. Furthermore, at the moment the enzyme halts, the 5–6 nucleotides between the junction and the halt point will be single-stranded and inside the enzyme, suggesting a different RNA structure-based mechanism of halting Xrn1. Although it is possible that these divergent secondary structures and sequences may possess three-dimensional folds or topologies similar to those from the MBFV, it is premature to make this prediction; it is just as possible that they use a different tertiary structure and strategy to block Xrn1. These observations suggest there are at least two secondary structural classes of xrRNAs within the flavivirus genus: the well-characterized class typified by the MBFVs and ISFVs (Class 1) and the class found in the TBFV and some of the NKVFVs (Class 2) (Fig. 6a).

It is also worth considering that outside the flavivirus genus but within the *Flaviviridae* family, the hepatitis C virus (a hepacivirus) and bovine viral diarrhea virus (a pestivirus) have been shown to block Xrn1 progression past a point proximal to the 5′-end of the viral genomic RNA, resulting in dysregulation of host RNA decay pathways[40]. However, Xrn1 resistance does not appear to be as robust as in the MBFVs and serves a somewhat different purpose in that it does not result in accumulation of a noncoding RNA that alters multiple host pathways during infection. Also, there is no evidence of a three-dimensional RNA structure that is similar to the xrRNAs found in the MBFV 3′-UTRs. Nonetheless, the presence of the ability to resist Xrn1 by these members of the *Flaviviridae* family outside the flavivirus genus show how useful this ability may be for viruses, inviting speculation that this function may exist throughout the viral RNA world. Indeed, exoribonuclease-resistant RNA sequences have been reported in some plant-infecting viruses that show no sequence similarity to the flavivirus xrRNAs[41].

It is informative to examine the diverse RNA structure-based ways to halt Xrn1 in light of the phylogenetic relationship of the *Flaviviridae* and their use of different arthropod vectors (Fig. 6b). In the viruses that are known to infect mosquitos (two groups of MBFVs and ISFVs), there is clear evidence for the existence of xrRNAs that follow a specific secondary structure pattern and adopt a similar three-dimensional fold. For the purposes of

discussion here, we refer to these as "Class 1" flavivirus xrRNAs. At least some likely NKVFVs use xrRNAs of this type, for example YOKV[25]. In contrast, the viruses that are known to infect ticks are using "Class 2" flavivirus xrRNAs with different secondary (and perhaps tertiary) structural features. Consistent with this, MODV and MMLV are members of the NKVFVs that are overall more closely related to TBFVs, and their xrRNAs match that class. Hence, the point of divergence of Class 1 from Class 2 xrRNAs appears to be near the point where MBFVs and TBFVs phylogenetically diverge and where the diverse NKVFVs lay. We hesitate to make strong conclusions based on these observations; nonetheless it will be interesting to see if future identification of any arthropod vectors for these various NKVFVs shows that the class of xrRNAs correlates with the vector; that is, if Class 1 xrRNAs are exclusive to mosquitoes and Class 2 xrRNAs are exclusive to ticks.

Within the areas of RNA-based nanotechnology, synthetic biology, and RNA-based therapies, viral RNAs such as xrRNAs can provide elegant building blocks for diverse applications. One can imagine a wide range of applications where it would be advantageous to control the degradation of specific RNA species. Given that the flavivirus xrRNAs are active against a range of exoribonucleases from different species, they might also be utilized as tools within a range of organisms to protect specific RNAs from 5′–3′ decay. The widespread presence of xrRNAs in the flaviviruses, the diversity of sequences and structures capable of mechanically blocking different exoribonucleases, and the apparent usefulness of this strategy for viral infection suggests that RNA structures capable of interfering with exoribonuclease activity may be more widespread than is currently appreciated.

## Methods

**Protein purification**. The RppH and Xrn1 enzymes were expressed and purified as described[22]. The RNase J1 expression vector was a kind gift of Ciaran Condon, and the Dxo1 expression vector was a kind gift of Liang Tong. RNase J1 was expressed in BL21 (DE3) cells in LB containing kanamycin and chloramphenicol at 30 °C. Expression was induced with 0.1 mM IPTG at OD600 = 0.6 and protein was expressed at 30 °C for 3 h. The cells were harvested by centrifugation at 5488×g for 10 minutes and the cell pellets were stored at −80 °C until further use. Dxo1 was expressed in Rosetta (DE3) cells in LB with kanamycin. The cells were grown at 37 °C to OD600 = 0.6–0.9. Expression was induced with 0.25 mM IPTG, the temperature was lowered to 16 °C and culture continued overnight before harvesting and storing similar to RNase J1 above. Cell pellets were lysed by sonication for 2 minutes processing time and centrifuged at 31,000×g for 30 minutes to clear the lysate. The protein was purified using Ni-NTA resin (Thermo) in a gravity flow column followed by size exclusion chromatography with either a Superdex 75 or Superdex 200 column. The final product was stored in buffer containing 20 mM Tris pH 7.3, 300 mM NaCl, 1 mM DTT or 2 mM BME, and 10% glycerol (1 mM EDTA was added to the RNase J1 sample) at −80 °C. The purity was assessed by SDS-PAGE and Coomassie staining.

**RNA synthesis**. Plasmids encoding the RNA sequences of interest under control of a T7 promoter were generated using standard molecular biology techniques. In brief, gene fragments (gBlocks) synthesized by Integrated DNA Technologies (IDT) were ligated into pUC19 vector (NEB) between the BamHI and EcoRI sites. Resultant plasmids were amplified in *Escherichia coli* (DH5α) and their sequences verified. To generate DNA template for transcription reactions, the desired region of the plasmid was amplified by PCR, then used in 5 mL in vitro transcription reactions and purified as previously described[22].

**Degradation resistance assay of the 3′-UTRs**. The exoribonuclease resistance assays were carried out by first folding 1–2 μg of 3′-UTR RNA and 1 μg of the 24-mer control RNA in a buffer composed of 50 mM Tris, pH 7.9, 100 mM NaCl, 10 mM MgCl₂, 1 mM DTT. This solution was incubated at 85 °C for 3 min, followed by 20 °C for 5 min, then held at 4 °C. Then, 0.8 μg of His-Xrn1 and 0.08 μg of His-BdRppH were added and the mixture was incubated at 37 °C for 2 h. The resulting RNA products were analyzed by denaturing PAGE[22].

**Mapping halt sites of 3′-UTRs**. Exoribonuclease halt sites were mapped by gel purifying His-Xrn1 digested 3′-UTR flavivirus RNA using the same protocol as mentioned above but at a larger scale (20 μg RNA total). Then RT-PCR was

performed using unique 5′-end-labeled primers for each flavivirus and GoScript Reverse Transcriptase (Promega)[22]. The RT products were resolved on a 12% denaturing polyacrylamide sequencing gel. The gels were visualized by phosphorimaging using a Typhoon 9400 Imager (Molecular Dynamics) and visualized with ImageQuant software.

**Chemical probing and computational analysis**. We utilized the one-dimensional (1D) chemical probing method described by Cordero et al. to conduct our SHAPE experiments and analysis[42]. In brief, RNAs were transcribed as described above, and purified using Agencourt AMPure XP magnetic beads. To anneal the RNA, it was heated to 90 °C for 2 minutes, transferred to ice for 2 minutes, and then placed at room temperature. RNA was modified via exposure to 24 mg/mL NMIA for 30 minutes before halting the reaction with an acid quench solution (1 volume 5 M NaCl, 1 volume 2 M HCl, and 1.5 volumes 3 M sodium acetate pH 5.2)[42]. Modified RNAs were then reverse transcribed at 42 °C for 45 minutes with a universal fluorescently labeled primer (IDT) (/5–6FAM/ AAAAAAAAAAAAAAAAAAAA GTTGTTGTTGTTGTTTCTTT). Labeled cDNA products were eluted in HiDi formamide and Gene Scan ROX 350 (Thermo) for capillary electrophoresis analysis on an Applied Biosystems 3500 XL Genetic Analyzer[42]. All reactions were performed in triplicate, as were "no modification" control reactions and ddNTP ladders. Analysis of capillary electrophoresis data was performed using the HiTRACE MATLAB toolkit (MathWorks) and 1D analysis pipeline developed by the Das and Yoon labs[43–46]. For further information, the website and tutorial for these programs and protocols can be found at: https://hitrace.github.io/HiTRACE/.

**Cell culture and viral infections**. The origin and culture conditions of the BHK-21J (obtained from the Laboratory of Professor C.M. Rice, at the time at Department of Molecular Microbiology, Washington University School of Medicine, St. Louis, Missouri) and C6/36 (ATCC CRL-1660) cells have been described[47–49]. Infections were performed as previously described[48]. Total RNA was isolated with Trizol (Invitrogen) at 30 h.p.i. from BHK-21J cells infected with MODV, APOIV, MMLV, or RBV (viruses were obtained from Professor J. Neyts at The Laboratory of Virology in the Rega Institute for Medical Research, Leuven, Belgium), or at 36 h.p.i. from CFAV-infected C6/36 cells (obtained from Professor X. Delamballerie at the UVE, Université Aix-Marseille II, Faculté de Médecine, Marseille, France).

**Replicon construction and transfection**. Viral RNA isolated from infected cells (see above) was dissolved in 30 ml H₂O and 5 μg was used for RT-PCR to amplify the Xrn1 stalling site-containing region of either MODV (nts. 10247–10343) or MMLV (nts. 10265–10367). Primers for these reactions can be found in Supplementary Table 1. The amplified MODV and MMLV cDNA fragments were cloned into Mlu I—Sph I-digested pSinrep5 and used for in vitro RNA transcription as described[26,50] (Supplementary Table 2).

**RNA transfection and northern blot analysis**. BHK-21J cells were transfected with 5 μg of in vitro transcribed Sinrep5 recombinant RNAs as described[48,50]. Approximately $1.5 \times 10^6$ transfected BHK-21J cells were seeded in a 35 mm plate. Total RNA was isolated from the transfected cells at 8 h post-electroporation using Trizol (Invitrogen). For Northern blotting, 7.5–10 μg of total RNA isolated from either infected or electroporated cells was denatured using formaldehyde and separated on a formaldehyde-containing 1.5% agarose gel and blotted to Hybond-N⁺ (GE Healthcare)[51]. The blots were hybridized using ³²P-labeled oligonucleotides as probes (Supplementary Table 1).

**Sequence Alignments**. Sequences were aligned using CLUSTALW though the online server. Accession numbers of sequences are contained in the appropriate figure legends.

**Data availability**. All data are available from the authors upon request.

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

## Acknowledgements

The authors thank current and former Kieft Lab members for thoughtful discussions and technical assistance. The expression vector for RNase J1 was a gift of Ciaran Condon, the expression vectors for Dxo1 and Xrn1 were gifts of Liang Tong, and the expression vector for RppH was a gift of Joel Belasco. This work was supported by NIH grant R35GM118070 to J.S.K., NIH training grant T32AI052066 to Z.O., and NIH K99GM115757 to E.G.C. J.S.K. was an Early Career Scientist of the Howard Hughes Medical Institute.

## Author contributions

Z.O'D., A.M., E.G.C., and J.S.K conceived of the study and designed the experiments. E.G.C. purified some of the enzymes and designed the primary assays. Z.O.D. and A.M. conducted the experiments. Z.O.D., A.M., and J.S.K. interpreted the data and wrote the manuscript with input from all authors. G.P.P. and M.G.S. generated initial sequence

alignments and some preliminary secondary structure predictions. P.A.G.C.S. and P.J.B. conducted viral infections and replicon transfections and subsequent Northern blots. R.C.O. contributed RNA folding and structure analysis.

## Additional information

**Competing interests:** The authors declare no competing financial interests.

