## [Peer Review File · Nature Communications]

Reviewers' comments:

Reviewer #1 (Remarks to the Author):

This manuscript represents (i) a mechanistic investigation of how 5'-to-3' exoribonuclease-blocking RNA structures cause stalling and (ii) a survey to identify additional functional RNA elements in related viruses. Overall, most of the data are very solid and their corresponding conclusions valid. The discussion is relevant, insightful, and comprehensive. The work is of interest to a wide audience from RNA specialists to virologists. Suggestions for improvement are provided below.

Fig2a – the nature of doublet band labeled “resistant product” should be explained – one would predict only a single xrRNA to be generated from a single xrRNA element present in the test RNA

Fig2d – The diagram is confusing for the initial + construct, and the last 4 constructs in the list. The cartoon implies that the reading frame for FLUC (indicted by standard notation, a rectangle) ends at the 3'-end of the FLUC reading frame, when it does not. For these constructs, I would suggest representing the single large ORF as a single rectangle and then coloring/shading in the FLUC and RLUC regions within the rectangle to indicate their positions within the ORF. To accommodate the xrRNA motifs, the height of the rectangles would need to be increased.

Ln 152: Change “progress through the intervening sequence” to “translate the intervening sequence” – so that readers are clear that the intervening sequence actively translated.

Have you tested if the xrRNAs can block scanning of 43S in the 5'UTR of capped messages?

Ln142 & 163: change “50%” to “40%”

Fig2d,e Rather than referring to a manuscript in prep. (ln151), evidence should be provided for the formation of the xrRNAs within the reporter mRNA context that was tested in this study.

Ln202-211 – these conjectures of structural equivalence should be tested/confirmed experimentally via point mutations/substitutions/compensatory mutants, etc.

Ln 238 delete “rigorous” wrt SHAPE – more compelling is the conservation of the overall structures with covarying base pairs (Fig5a,b).

Fig5 – has two panel c's

Fig5 & ln278 – The putative long-range interaction should be further tested with compensatory mutations that target single and double base pairs in the predicted 4bp interaction. For example, change the central GC pair only, the central AU pair only, and then both simultaneously. This approach is generally more effective at recovering at least one

positive set of compensatory mutants, because (i) the disruption of one bp (or 2bps) in a set of four is usually sufficient to effectively disrupt the interaction (ii) the minimal substitutions reduces the chance of inducing alternative non-functional structures and (iii) one gets several chances for identifying a functional set of compensatory mutants. Another good approach is to introduce compensatory mutations that correspond to covariation that exists naturally in other variants of the virus.

Ln333 – can the Authors' provide any additional insights as to what could be responsible for the multiple stall sites for Dox1.

Ln 353 – The halt point maps to a helix in the predicted secondary structure, however this helix would presumably be separated/digested by the enzyme's ssRNA-specific entrance tunnel/active site. Therefore, pointing out the dsRNA nature of the substrate RNA with respect to the stall site does not seem very relevant. Instead, although the lower stem is not part of the xrRNA, it could contribute to the efficient formation/stabilization of the downstream structure that does the blocking – a point that may be worth mentioning.

Reviewer #2 (Remarks to the Author):

Infection with different flaviviruses is associated to the accumulation of viral non-coding RNAs, called subgenomic flavivirus RNAs or sfRNAs, which are the product of incomplete degradation of the viral genome by the host exoribonuclease Xrn1. These sfRNAs play multiple function that are relevant for infection. In present manuscript, the authors study mechanistic aspects by which viral RNA structures, named xrRNAs, function as structural blocks for Xrn1 progression, resulting in formation of sfRNAs. Interestingly, the authors shown that a West Nile virus xrRNA is able to halt the activity of two exonucleases unrelated to Xrn1, something that has not been previously tested. In addition, they extend the analysis of RNA structures present at the 3'UTRs of different flavivirus groups. The flaviviruses can be ecologically divided into four groups, mosquito borne (MBFVs), tick borne (TBFVs), insect specific (ISFVs) and flaviviruses with no known vector (NKVFs), which replicate only in vertebrate hosts. Here, the authors explore the presence of xrRNAs in different flavivirus groups and propose the existence of different classes of xrRNAs (one associated to viruses that infect mosquitos and the other class associated to viruses that infect ticks). Overall, the manuscript provides observations that extend original studies by the same authors. However, there are important issues that reduce the impact of the findings and my enthusiasm about the work.

First, authors indicate that they provide new mechanistic information underlying the Xrn1 resistance by MBFVs xrRNAs, however, the conclusions are similar to the ones already reported by the same authors (references 23, 27).

Second, authors identified new xrRNAs from different flaviviruses and grouped them into two classes, however, the grounds for this classification are not well supported. In this regard, limited structural information is provided to support the classification. It is important

to mention that the authors have previously shown how difficult is to predict xrRNA structures and only with crystallographic data was possible to define critical unpredicted base pairs, three way interactions, ring like structures. Thus, it is surprising that they make conclusions about similar structural properties of insect specific and MBFV xrRNAs, which are not related in sequence/structures (Figure 3, prediction of non-canonical base pairs and other contacts). Because of the complexity of the RNA structures present in the 3'UTRs of different groups of flaviviruses, classification of xrRNAs should be done more rigorously with better structural and functional information.

In addition, there are statements that are biased to accommodate the proposed classification. For instance, line 216, "the NKVFs do not have sequences that match the MBFV or ISFV xrRNAs". The NKVFs are not a monophyletic group. There is a subgroup of NKVFs related to MBFVs (such as YOKV) and other group related to TBFVs (such as MODV), therefore, depending on the viruses chosen they would share or not structural features with members of the two ecological groups.

Other comments

Line 239 "the secondary structures of these divergent xrRNAs contain a three-way junction, but one that differs dramatically from those found in the MBFV and ISFV". It is difficult to support this idea without structural information.

Figure 5 there is a mistake (labeling of the panels)

Response to Reviewers: MacFadden and O'Donoghue et al.

We would like to thank the referees for their time and careful consideration of our manuscript. We were pleased to read that there was general enthusiasm expressed by the referees in terms of the topic, the quality of the work, the impact, and the general presentation. The referees also pointed out several areas of concern and places where the manuscript could be improved. We carefully considered the comments and have revised the manuscript to reflect them. We also conducted several more experiments to address key concerns. We are confident that we have addressed all of the referees' criticisms and that the manuscript is greatly improved. Below, we present the full text of the referees' comments, our responses, and how we have altered the manuscript.

Reviewer #1 (Remarks to the Author):

This manuscript represents (i) a mechanistic investigation of how 5'-to-3' exoribonuclease-blocking RNA structures cause stalling and (ii) a survey to identify additional functional RNA elements in related viruses. Overall, most of the data are very solid and their corresponding conclusions valid. The discussion is relevant, insightful, and comprehensive. The work is of interest to a wide audience from RNA specialists to virologists. Suggestions for improvement are provided below.

Fig2a – the nature of doublet band labeled “resistant product” should be explained – one would predict only a single xrRNA to be generated from a single xrRNA element present in the test RNA

Discussion: The doublet effect was likely due to 3' heterogeneity of the input RNA or a gel artifact.

Specific revision: We have replaced that panel with a better gel.

Fig2d – The diagram is confusing for the initial + construct, and the last 4 constructs in the list. The cartoon implies that the reading frame for FLUC (indicted by standard notation, a rectangle) ends at the 3'-end of the FLUC reading frame, when it does not. For these constructs, I would suggest representing the single large ORF as a single rectangle and then coloring/shading in the FLUC and RLUC regions within the rectangle to indicate their positions within the ORF. To accommodate the xrRNA motifs, the height of the rectangles would need to be increased.

Ln 152: Change “progress through the intervening sequence” to “translate the intervening sequence” – so that readers are clear that the intervening sequence actively translated.

Have you tested if the xrRNAs can block scanning of 43S in the 5'UTR of capped messages?

Ln142 & 163: change “50%” to “40%”

Fig2d,e Rather than referring to a manuscript in prep. (ln151), evidence should be provided for the formation of the xrRNAs within the reporter mRNA context that was tested in this study.

Discussion: These concerns are all in regard to Figure 2d&e, which contains evidence that an elongating ribosome can progress through xrRNA structures. We agree that there are some controls and additional analysis that are important to complete this experiment. However, these additional analyses may dilute the main thrust of the manuscript, and are likely better presented in another study.

Specific revision: We have removed these data from the manuscript so that the main focus is not lost; we believe this improves the manuscript. The data will be presented in a future publication along with validation of xrRNA function in yeast and other analysis.

Ln202-211 – these conjectures of structural equivalence should be tested/confirmed experimentally via point mutations/substitutions/compensatory mutants, etc.

Discussion: We agree that assessing the ability of the CFAV xrRNAs to assume the same tertiary interactions as are observed in previously characterized xrRNAs is an important test for our predictions. This is especially true in the cases where the sequence has changed but we predict equivalent interactions can occur (e.g., the base triple).

Specific revision: We have made and tested several CFAV xrRNA mutants. The results support the idea that these RNAs are using a very similar set of tertiary interactions to the MBFVs. These results are presented in a new figure panel (Fig. 3g) and in the main text.

Ln 238 delete “rigorous” wrt SHAPE – more compelling is the conservation of the overall structures with covarying base pairs (Fig5a,b).

Discussion: We agree that the word “rigorous” is a poor choice, and that the use of conservation along with the chemical probing is critical.

Specific revision: We have revised the text to reflect this point.

Fig5 – has two panel c’s

Specific revision: We apologize for this typographic error. It has been corrected.

Fig5 & Ln278 – The putative long-range interaction should be further tested with compensatory mutations that target single and double base pairs in the predicted 4bp interaction. For example, change the central GC pair only, the central AU pair only, and then both simultaneously. This approach is generally more effective at recovering at least one positive set of compensatory mutants, because (i) the disruption of one bp (or 2bps) in a set of four is usually sufficient to effectively disrupt the interaction (ii) the minimal substitutions reduces the chance of inducing alternative non-functional structures and (iii) one gets several chances for identifying a functional set of compensatory mutants. Another good approach is to introduce compensatory mutations that correspond to covariation that exists naturally in other variants of the virus.

Discussion: This is an excellent suggestion. We generated a new set of mutants in which we introduced less “aggressive” changes to the sequences comprising the putative pseudoknot interaction. Resistance assays with these new mutants clearly establish the presence of a pseudoknot interaction, as each individual mutation disrupts resistance, but the double mutation (compensatory) restores resistance.

Specific revision: We have replaced the diagram & data in Figure 5c with these new results and changed the text accordingly.

Ln333 – can the Authors’ provide any additional insights as to what could be responsible for the multiple stall sites for Ddx1.

Discussion: The “stuttering” that we observe with Ddx1 is interesting. As we state in the manuscript, this provides evidence for the idea that the physical interaction of the RNA with the surface of the enzyme, and thus the structure of the enzyme around the active site, matters for xrRNA function. However, we hesitate to make speculation beyond this without more detailed structural data (such as currently unavailable structures of xrRNA-enzyme complexes).

Specific revision: We have modified the text to ensure that potential implications of this observation are clear, but we avoid too much speculation.

Ln 353 – The halt point maps to a helix in the predicted secondary structure, however this helix would presumably be separated/digested by the enzyme’s ssRNA-specific entrance tunnel/active site. Therefore, pointing out the dsRNA nature of the substrate RNA with respect to the stall site does not seem very relevant. Instead, although the lower stem is not part of the xrRNA, it could contribute to

the efficient formation/stabilization of the downstream structure that does the blocking – a point that may be worth mentioning.

Discussion: This is an insightful observation, and one that we also noticed. We did not discuss this point in the initial submission, as we did not feel that we understood the full implications. However, as the reviewer also took note of this, we feel more confident in discussing it and its implications.

Specific revision: We have added text to the discussion to address this point.

Reviewer #2 (Remarks to the Author):

Infection with different flaviviruses is associated to the accumulation of viral non-coding RNAs, called subgenomic flavivirus RNAs or sfRNAs, which are the product of incomplete degradation of the viral genome by the host exoribonuclease Xrn1. These sfRNAs play multiple function that are relevant for infection. In present manuscript, the authors study mechanistic aspects by which viral RNA structures, named xrRNAs, function as structural blocks for Xrn1 progression, resulting in formation of sfRNAs. Interestingly, the authors shown that a West Nile virus xrRNA is able to halt the activity of two exonucleases unrelated to Xrn1, something that has not been previously tested. In addition, they extend the analysis of RNA structures present at the 3'UTRs of different flavivirus groups. The flaviviruses can be ecologically divided into four groups, mosquito borne (MBFVs), tick borne (TBFVs), insect specific (ISFVs) and flaviviruses with no known vector (NKVFs), which replicate only in vertebrate hosts. Here, the authors explore the presence of xrRNAs in different flavivirus groups and propose the existence of different classes of xrRNAs (one associated to viruses that infect mosquitos and the other class associated to viruses that infect ticks). Overall, the manuscript provides observations that extend original studies by the same authors. However, there are important issues that reduce the impact of the findings and my enthusiasm about the work.

First, authors indicate that they provide new mechanistic information underlying the Xrn1 resistance by MBFVs xrRNAs, however, the conclusions are similar to the ones already reported by the same authors (references 23, 27).

Discussion: We disagree with this point. In the two publications noted, we presented the structure of two mosquito-borne flavivirus (Murray Valley encephalitis and Zika) xrRNAs, with accompanying functional data and analyses. Based on the data, we presented models, or hypotheses regarding the mechanism of resistance; we did not make conclusions. However, these ideas were not tested and thus critical aspects of the mechanism were unknown. This includes the need for specific RNA-protein interactions, the idea that the interaction with the RNA and the enzyme's surface was important, the generality of exonuclease resistance, etc. The work presented here is thus an essential next step in understanding the mechanism of xrRNAs.

Specific revision: We have added text to the discussion to more clearly articulate the motivation for these experiments and their importance to understanding xrRNAs, as well as implications for the use of xrRNAs as experimental or synthetic biology tools.

Second, authors identified new xrRNAs from different flaviviruses and grouped them into two classes, however, the grounds for this classification are not well supported. In this regard, limited structural information is provided to support the classification. It is important to mention that the authors have previously shown how difficult is to predict xrRNA structures and only with crystallographic data was possible to define critical unpredicted base pairs, three way interactions, ring like structures. Thus, it is surprising that they make conclusions about similar structural properties of insect specific and MBFV xrRNAs, which are not related in sequence/structures (Figure

3, prediction of non-canonical base pairs and other contacts). Because of the complexity of the RNA structures present in the 3'UTRs of different groups of flaviviruses, classification of xrRNAs should be done more rigorously with better structural and functional information.

Discussion: This comment appears to be due to some misunderstanding of the grounds and purpose of the classification. The proposed classification is based on clear differences in the secondary structure patterns observed, and makes no prediction about differences or similarities beyond that. Clear differences in the secondary structures include: the configuration of the 3-way junction, the length of intervening sequence between the junction and the pseudoknot interaction, and the stop point of the enzyme relative to other features. Our classification does not make any assumptions about what the 3-D fold will look like, it only takes into account the secondary structure.

Within the 3-D fold there may indeed be similarities between class 1 and class 2; our classification does not preclude this. Such an observation would be interesting, as it would mean that two very different secondary structure patterns could give rise to a similar fold. Within such a structure, certainly we will observe unpredicted base pairs, triple interactions, base and helical stacking, and other noncanonical interactions. However, the specific interactions that we have observed in the “class 1” are likely to be different in the “class 2,” given the sequence and secondary structure variation.

Finally, we point out that similar classifications are common in RNA structure analysis and have proved very useful in organizing experiments and discussion of these other RNAs. Even when the 3-D structures turn out to be similar, assigning classifications to the secondary structures is useful. This includes RNAs such as internal ribosome entry sites (e.g., the *Dicistroviridae* type 1 and type 2 IRESs), riboswitches, etc.

Specific revision: We have added text to the discussion to more clearly articulate the basis for the classification, and also point out that such classification is common and useful in RNA structure discussions.

In addition, there are statements that are biased to accommodate the proposed classification. For instance, line 216, “the NKVFs do not have sequences that match the MBFV or ISFV xrRNAs”. The NKVFs are not a monophyletic group. There is a subgroup of NKVFs related to MBFVs (such as YOKV) and other group related to TBFVs (such as MODV), therefore, depending on the viruses chosen they would share or not structural features with members of the two ecological groups.

Discussion: We apologize if our language appears biased. We were perhaps imprecise in regard to how we presented the discussion about the NKVFs. However, we point out that indeed we presented the NKVFs in two groups (not as a monophyletic group) in the diagram of Figure 6b, where we include YOKV in one group and other NKVFs in another group. This was based on previous classification in the literature, and we find it compelling that the RNA secondary structures independently correlate with this phylogenetic classification. Thus we fully agree that the NKVFs are diverse in terms of their xrRNA types; this does not alter the fact that these xrRNAs thus far fall into two secondary structural classes.

Specific revision: We have adjusted the text to better present the nature of the NKVFs as a group and how this related to the secondary structure patterns we observe.

Other comments

Line 239 “the secondary structures of these divergent xrRNAs contain a three-way junction, but one that differs dramatically from those found in the MBFV and ISFV”. It is difficult to support this idea without structural information.

Discussion: We disagree. The secondary structures of the three-way junction are clearly different, for which we have good structural information; we make no claims to similarities or differences in 3-D structures. Indeed, the helical stacking configurations in the two types could ultimately turn out to be the same, but the secondary structures that give rise to these are clearly different. That would be an intriguing result, should it turn out to be true.

However, we point out that detailed analyses of RNA three-way junctions and their different helical stacking arrangements have been cataloged (ref. 37 and follow-on studies). These studies allow the configuration of many three-way junctions to be predicted based solely on secondary structure information. In the case of class 1, the configuration of the 3-way junction was correctly predicted based on these analyses (family C). Class IIs are very difficult to place in a specific 3-way junction family, hence we avoided doing so. However, the secondary structure patterns are clearly different.

Specific revision: We have adjusted the text to ensure the reader is not confused as to the difference between secondary structure and 3-D folding.

Figure 5 there is a mistake (labeling of the panels)

Specific revision: We apologize for this typographic error. It has been corrected.

REVIEWERS' COMMENTS:

Reviewer #1 (Remarks to the Author):

This Reviewer is satisfied with the responses and actions taken by the Authors. The revised manuscript is now suitable for publication.

Reviewer #2 (Remarks to the Author):

This is a revised version of a manuscript that investigates the process of flavivirus xRNA resistance to exonuclease activity. The work extends previous studies of the same authors. The manuscript adds new information regarding: 1. the generality of exonuclease resistance: authors demonstrate that xRNA not only stops the activity of Xrn1 but also the activities of other exonucleases. These in vitro experiments using bacterial and yeast enzymes provide new information that supports the original model, in which it was proposed that the xRNAs function as mechanical blocks. These experiments support the idea that there are not specific enzyme-RNA contacts. 2. The function of xRNAs previously predicted in ISFV and NKFV were experimentally confirmed.

In my opinion, the new version of the manuscript addresses most of the concerns raised by the reviewers and includes data of mutated RNAs (CFAV and MODV) that straightened their original conclusions.